# Identification of Signatures of Selection for Litter Size and Pubertal Initiation in Two Sheep Populations

**DOI:** 10.3390/ani12192520

**Published:** 2022-09-21

**Authors:** Zhishuai Zhang, Zhiyuan Sui, Jihu Zhang, Qingjin Li, Yongjie Zhang, Chenguang Wang, Xiaojun Li, Feng Xing

**Affiliations:** Key Laboratory of Tarim Animal Husbandry Science and Technology, Xinjang Production and Construction Group, College of Animal Science, Tarim University, Alaer 843300, China

**Keywords:** Hetian sheep, Cele Black sheep, reduced-representation genome sequencing, fecundity, genetic diversity, SNPs

## Abstract

**Simple Summary:**

Fecundity is an important economic trait of sheep that directly affects their economic and productive efficiency. Our study aimed to identify single nucleotide polymorphism (SNP) loci associated with sheep puberty or litter size which could be used in future breeding programs to improve fertility. We obtained one SNP on GNAQ gene (NC_040253.1: g.62677376G > A), which was associated with litter size in Cele Black sheep. Perhaps this SNP could be a criterion to screen polytocous sheep.

**Abstract:**

Fecundity is an important economic trait in sheep that directly affects their economic and productive efficiency. Our study aimed to identify SNP loci associated with sheep puberty or litter size which could be used in future breeding programs to improve fertility. Genomic DNA was obtained from Hetian and Cele Black sheep breeds and used for reduced-representation genome sequencing to identify SNP loci associated with pubertal initiation and litter size. Selective signatures analysis was performed based on the fixation index and nucleotide diversity, followed by pathway analysis of the genes contained in the selected regions. The selected SNP loci in the genes associated with pubertal initiation and litter size were validated using both sheep breeds. In total, 384,718 high quality SNPs were obtained and 376 genes were selected. Functional annotation of genes and enrichment analysis identified 12 genes associated with pubertal initiation and 11 genes associated with litter size. SNP locus validation showed that two SNP on PAK1 and four on ADCY1 may be associated with pubertal initiation, and one SNP on GNAQ gene (NC_040253.1: g.62677376G > A) was associated with litter size in Cele Black sheep. Our results provide new theoretical support for sheep breeding.

## 1. Introduction

Puberty is a critical stage, when animals first acquire fertility and involves complex interactions [1,2,3]. In animal production, breeding females with early menarche can reduce feeding costs and increase lifetime production, therefore shortening the generation interval and accelerating the breeding process [4,5,6]. The polytocous traits of sheep are also influenced by genetic, hormonal, environmental, and nutritional factors [7,8,9]. Interest in the research and identification of proteins associated with these traits is increasing because they have potential for use as protein biomarkers for polytocous traits, which would be extremely beneficial for sheep breeding.

Selection, both artificial and natural, has generated the many sheep breeds existing worldwide. During selection, the frequency of certain alleles within the population increases and the surrounding associated chromosomal regions show a decrease in polymorphism due to the hitchhiking effect, a phenomenon known as selective signatures [10]. Sequencing technology to study trait-associated genes has been widely utilized [11,12]. Wei et al. [13] used Fst and XP-EHH methods on 122 Chinese native sheep and found that EPASI genes play an important role in hypoxia adaptation. By selective signature analysis between polytocous and monotocous sheep populations, Wang et al. identified six genes (ESR1, OXTR, MAPK1, RYR1, PDIA4, and CYP19A1) associated with polytocous traits in sheep [14]. Yao et al. used selective signature analysis between high and low fertility populations in Bamei mutton sheep and found that JUN, ITPR3, PLCB2, HERC5 and KDM4B genes might be associated with polytocous traits [15]. Li et al. [16] identified the PDGFD gene as a possible important candidate gene affecting tail fat deposition in sheep by selective signatures analysis. Xin et al. [17] identified the P4-Ins-13bp mutation of the IGF2BP2 gene in goats as a potential molecular marker significantly associated with lambing traits by selective signatures analysis. Wang et al. conducted a selection elimination analysis on Jining gray goats with one, two, and three firstborn litters and found that KIT, KCNH7, KMT2E, PAK1, PRKAA1, and SMAD9 genes may regulate polytocous traits in this sheep [18].

The Cele Black Sheep and the Tian Sheep are two local breeds in Northwest China, characterized by high and low fertility, respectively [19]. Hetian sheep is a wool-producing breed that lives in the Hotan region of Xinjiang. It has characteristics of heat resistance, rough processing resistance and disease resistance [20]. In Xinjiang, Cele Black sheep produce more lambs per litter and have more estrus cycles in a year than other breeds. Although the environment in Xinjiang is very harsh, with hot summers, cold winters and windy and dry conditions all year round, ewes of this breed produce 2–4 lambs per litter [21]. Hetian sheep reach puberty at approximately 8–12 months of age and breed at 18–24 months of age, with an average litter size of 102.52% [22]. The puberty of Cele Black sheep generally occurs earlier than in other sheep breeds. Ewes become sexually mature at 4–5 months of age and breed after one year of age, with an average litter size rate of 215% [23].

Sheep are one of the most important domestic animals in Xinjiang. Improving the productivity of sheep plays an important role in improving the quality of life of local people and is one of the important goals of sheep breeding. However, the low fertility of most domesticated sheep is one of the factors limiting the development of sheep breeding in Xinjiang [19]. In this study, we performed selective signatures analysis based on the differences in pubertal initiation time and litter size between Hetian and Cele Black sheep to obtain single nucleotide polymorphism (SNP) loci associated with these two traits. This can provide theoretical support for the selection and breeding of sheep in Xinjiang.

## 2. Materials and Methods

### 2.1. Sample Cohort and DNA Extraction

A total of 342 blood samples from a natural population of healthy ewes between the ages of three and four years were collected and stored at −80 °C until use. The exploratory cohort for selection signatures analysis comprised 54 Cele Black sheep (No litter size record; Group CL) and 26 Hetian sheep (No litter size record; Group HT). To validate the relationship of SNP loci with litter size we used a validation cohort of 77 monotocous Cele Black sheep (litter size = 1; Group CL-1), 68 polytocous Cele Black sheep (litter size = 2; Group CL-2). These 145 Cele Black sheep (CL-1 and CL-2) and 117 Hetian sheep (Group HT-1) were used as a cohort for validating the relationship of SNP loci with puberty. Genomic DNA was extracted using the standard phenol/chloroform extraction method. DNA integrity was verified using a 1% agarose gel and the OD 260/280 ratio was obtained by NanoDrop™ spectrophotometer (Thermo Fisher Scientific, Waltham, MA, USA). The DNA concentration was measured using a Qubit (Thermo Fisher Scientific). This study was conducted in accordance with the specifications of the Ethics Committee of Tarim University of Science and Technology. All the sheep were collected from Xinjiang Kunlun Luyuan Sheep Farm in Cele County, Hotan City, Xinjiang Province.

### 2.2. Library Construction and Sequencing

Library construction was performed using the genotyping-by-sequencing (GBS) method. Genomic DNAs from CL and HT were digested with restriction endonucleases MseI and SacI (New England Biolabs, Ipswich, MA, USA) before being ligated with specific adaptors. An aliquot of the product was pooled and purified with AMPure XP Beads (Beckman Coulter, Brea, CA, USA). Polymerase chain reaction (PCR) enrichment was then performed using the high-fidelity polymerase KOD-Plus-Neo (Toyobo Co. Ltd., Osaka, Japan). All products were pooled and loaded onto an electrophoresis gel overnight with Certified Megabase Agarose (Bio-Rad, Hercules, CA, USA) at low pressure. Products in the range of 380–480 bp were purified using a gel extraction kit (QIAGEN, Hilden, Germany). Libraries were pooled according to the target downstream data volume and paired end 150 bp (PE150) sequencing was performed using the Illumina HiSeq platform (Illumina, San Diego, CA, USA). Each library contains 40 samples and we matched the clean reads individually to the barcodes and remnant restriction sites at both ends.

### 2.3. Quality Control, Comparison and Identification of SNP Loci

Quality control and data filtering were performed using fastp software [24] with the following filtering criteria: (1) Remove polyG and polyX at the end of the reads (minimum length of 10 bp). (2) Reads with N numbers greater than 5 were removed. (3) Remove reads with a base mass ratio below 15, higher than 40%. (4) Filtered reads with lengths below 15 were removed. The BWA package [25] version:0.7.17; parameter: mem) was then used to compare the filtered clean reads with the reference genome [26]. Variant loci were detected using GATK [27] (version 4.1.9.0; parameters: HaplotypeCaller, CombineGVFs, GenotypeGVFs). All variant loci were annotated using ANNOVAR [28] (version 20200607; default parameters). The SNP and insertion–deletion mutations (INDEL) loci were filtered separately using GATK. The following criteria were used for SNP filtering: QualByDepth (QD) < 2.0; Quality (QUAL) < 30.0; StrandOddsRatio (SOR) > 3.0; FisherStrand (FS) > 60.0; RMSMappingQuality (MQ) < 40.0; MappingQualityRankSumTest (MQRankSum) < −12.5; ReadPosRankSumTest (ReadPosRankSum) < −8.0. The following criteria were used for INDEL filtering: QD < 2.0; QUAL < 30.0; FS > 200.0; MQ < 40.0; ReadPosRankSum < −20.0. For the GATK-filtered loci, vcftools [29] were used to exclude loci with minor allele frequency (maf) less than 3% [30] and genotype deletion rate greater than 20%.

### 2.4. Genetic Diversity Analysis

The frequency, number of alleles, and nucleotide diversity (π) were calculated using vcftools software and the polymorphism information content (PIC) values of the SNP data were calculated using the following equation:$$\mathrm{PIC}=1-\overset{n}{\underset{i=1}{\sum }}P_{i}^{2}{{}_{\;}}^{\;}-\sum_{i=1}^{n-1}\sum_{j=j+1}^{n}2P_{i}{}^{2}P_{j}{}^{2}$$
where *n* is the number of alleles and *P_i_* and *P_j_* denote the ith and jth allele frequency, respectively.

Genetic diversity (GD) and observed heterozygosity (Ho) were calculated using PLINK [31]. We used PopLDdecay [32] with default parameters to plot linkage disequilibrium (LD) decay, based on the allele frequency correlation (r^2^) to reveal the population relationship between HT and CL. The average number of nucleotide differences (K) was calculated as the sum of nucleotide variations of all individuals divided by the number of individuals.

### 2.5. Population Structure Analysis

Neighbor-joining trees were constructed using PHYLIP [33] and data were visualized using ggtree [34]. Principal component analysis (PCA) was performed using GCTA [35] to obtain the principal component values of each sample. The R package scatterplot3d [36] was then used to plot the principal component scatter plots in order to further investigate the population genetic structure. ADMIXTURE [37] was used to infer the population structure.

### 2.6. Selective Signatures Analysis and Protein Interaction Analysis

Selective signature analysis was performed using HT as the reference and CL as the target population in order to identify genes associated with pubertal initiation and litter size. The differentiation index (Fst) and nucleotide diversity (π) were calculated by vcftools, parameter: —fst-window-size 100,000—fst-window-step 50,000, for each interval between the two breeds (100 KB as a window and 50 KB as a step to divide the genome). Plots were drawn using the R package cmplot. Using the methods described by Li et al. [38], regions with both extremely low or high θπ ratios (5% left-tailed and 95% right-tailed) and significantly high Fst values (Fst in the top 5%) were selected as genomic candidate region. Gene functions for the genes in selected regions were annotated via alignment with the Gene Ontology (GO) and Kyoto Encyclopedia of Genes and Genomes (KEGG) databases. Then we reviewed the references and screened for genes associated with sheep puberty or litter size. Protein interaction analysis of the screened genes was performed by STRING [39].

### 2.7. SNP Loci Validation

For SNP validation, we used CL-1, CL-2, and HT-1 groups. CL-1 and CL-2 were used to validate the relationship of SNP loci with litter size. CL-1, CL-2, and HT-1 groups were used to validate the relationship of SNP loci with puberty. SNP typing was performed by the Beijing Compass Biotechnology Company, China. The SNP loci located in screened genes that were associated with sheep puberty or litter size were validated for genotype-litter size correlation in CL-1 and CL-2 using Fisher’s exact test method [40]. The genotype-pubertal initiation correlation was validated between HT-1 and Cele Black (CL-1 and CL-2) populations.

## 3. Results

### 3.1. Reduced-Representation Genome Sequencing and Data Filtering

We obtained 41.859 G of clean data, 0.076–1.174 G per sample, with an average of 0.523 G. An average of 4.36 million reads were retained per sample among the clean reads; q20 was higher than 98.03%, q30 was higher than 92.92% and the GC content was stable between 45.305 and 47.734%. Overall, our sequencing data yielded a high-quality score and a total of 384,718 high-quality SNPs were identified.

### 3.2. Population Genetic Diversity

Table 1 shows the PIC, Ho, GD, nucleic acid diversity (π), and the average number of nucleotide differences (K) were all higher in HT than in CL, indicating that Hetian sheep had higher genetic diversity. Linkage disequilibrium (LD) analysis showed a small difference in the decay rate of LD coefficients between the HT and CL (Figure 1A). CL showed a faster decay rate (Figure 1B).

### 3.3. Population Structure Analysis

The population structure was assessed using CL and HT. Both the evolutionary tree and PCA results identified two distinct groups (Figure 2A,B). The structural analysis showed that the most likely value of K was 2 (Figure 2C,D). This is consistent with the PCA data and clustering of the two breeds and therefore indicates that CL and HT have two different ancestral genetic backgrounds.

### 3.4. Selective Signatures Analysis

Based on selective signatures analysis (Figure 3), we identified 376 genes under selection (Appendix A). Based on GO annotation, KEGG enrichment analysis and related references of the 376 genes, 12 genes (*PLCE1*, *SPIRE1*, *DIAPH2*, *GABRA3*, *GNAQ*, *GHR*, *DNMT3B*, *PLCB1*, *ABCC8*, *PTGFR*, *PAK1*, *ADCY1*) were associated with pubertal initiation and 11 genes (*ADCY1*, *POU1F1*, *BMPR2*, *HS6ST1*, *LHCGR*, *DNMT3B*, *GHR*, *LIF*, *PAK1*, *TGFB2*, *GNAQ*) were associated with litter size. From these genes, we selected 25 SNP loci with high Fst values for further validation. Protein interaction analysis revealed interactions between *PTGFR*, *LHCGR*, *GNAQ*, *GHR*, *PLCE1*, *LIF*, *PLCB1*, *ADCY1*, *BMPR2*, and *TGFB2* (Figure 4). The selected genes included several SNPs potentially linked with pubertal initiation: six SNP loci in *GNAQ*, three in *GHR*, one in *DNMT3B*, two in *PAK1*, and six in *ADCY1*. Furthermore, several SNPs linked with litter size were found: six SNP sites were located in *ADCY1*, one in *POU1F1*, four in BMPR2, one in *LHCGR*, one in *DNMT3B*, three in *GHR*, one in *LIF*, two in *PAK1*, and six in *GNAQ*.

### 3.5. SNP Loci Validation

We found that two SNP loci in *PAK1* (NC_040272.1: g. 20302507T > A and NC_040272.1: g. 20312315G > T) and four SNP loci in *ADCY1* (NC_040252.1: g. 127782749T > G, NC_040252.1: g. 127782799G > A, NC_040252.1: g. 127782815G > T and NC_040252.1: g. 127782942T > C) showed significant differences (*p* < 0.05) between HT-1 and the CL-1 and CL-2 groups. These SNPs were thought to be associated with primiparous initiation. Another SNP locus in *GNAQ* showed highly significant differences (*p* < 0.01) between CL-1 and CL-2, suggesting that this SNP locus was associated with litter size in Cele Black sheep. The heterozygous genotype GA at the NC_040253.1: g.62677376G > A locus on the *GNAQ* gene was more favorable for producing polytocous sheep (Appendix A).

## 4. Discussion

In this study, we characterized population genetic diversity and genomic selection in 54 CLs of high fertility and 26 HTs of low fertility. We assessed the population structure of these two sheep populations using NJ tree, ADMIXTURE and PCA. In addition, we performed Selection signatures analysis using FST and π methods to screen for SNP loci associated with puberty or little size. Finally, we validated the little size associated SNP loci using Celle black sheep (CL-1 and CL-2) with little size records and performed preliminary validation of SNP loci associated with puberty using Hetian sheep (HT-1) and Cele Black sheep (CL-1 and CL-2).

The assessment of genetic variation within breeds/populations can be an important basis in the improvement and development of sheep breeding [41,42]. Species with high genetic diversity have a greater ability to adapt to their environment [43]. In our study, genetic diversity of Hetian sheep was higher than that of Cele Black sheep. This indicates that Hetian sheep have higher adaptability to the environment. The decay of LD was influenced by the recombination rate and the number of recombination generations. CL showed a faster decay rate (Figure 1B) indicating an earlier initiation of puberty and higher litter size than HT (Figure 1C). This is consistent with previous published results [22,23].

To determine whether Hetian and Cele Black sheep populations are suitable to the selection signatures analysis, we conducted a population structure analysis. The resulting data showed that Hetian sheep and Cele Black sheep have different ancestral genetic backgrounds.

Selective signatures can reveal genomic regions subject to selection that are associated with phenotypic traits acquired during biological evolution [44,45]. In this study, we used reduced-representation genome sequencing technology to analyze Hetian and Cele Black sheep and applied selective signatures analysis to identify SNPs associated with pubertal initiation and litter size in sheep.

Selection signatures analysis identified two SNP loci in *PAK1* and four SNP loci in *ADCY1* potentially associated with pubertal initiation in sheep, and one SNP locus in *GNAQ* that was significantly associated with litter size in sheep. P21-activated kinase-1 (PAK1) is an important mediator of estrogen’s cell survival functions [46]. In addition, PRL and estrogen interact with each other through PAK1 [47]. Pak1 is able to phosphorylate the Ser305 site of estrogen receptor alpha (ER α), a modification that is important in the transactivation function of ER α [48]. Therefore, PAK1 may indirectly control the initiation of puberty in sheep by regulating estrogen. Expression of adenylyl cyclase (ADCY1) was reported to induce enrichment of cyclic adenosine monophosphate at the zona pellucida and nucleus of the oocytes and is a main second messenger in oocytes, where is involved in the regulation of the meiotic process [49]. Therefore, the *ADCY1* gene may promote the initiation of proestrus in sheep by controlling meiosis. Previous studies have shown that the GNAQ gene is significantly associated with little size in Kazakh, Chinese Merino and Hu sheep [50]. Our study adds to this point that the GNAQ gene was also significantly associated with little size in Cele Black sheep. The heterozygous genotype GA at the NC_040253.1: g.62677376G > A locus on the *GNAQ* gene was more favorable for producing polytocous sheep. However, the age at pubertal initiation for each sheep was not collected and therefore further validation of the identified loci associated with pubertal initiation is needed.

## 5. Conclusions

We identified two SNP loci on the *PAK1* gene and four SNP loci on the *ADCY1* gene that may be associated with pubertal initiation in sheep based on reduced-representation genome sequencing using selection elimination analysis. One SNP (NC_040253.1: g.62677376G > A) locus on the *GNAQ* gene was found to be associated with lambing number in sheep. Hetian sheep had higher genetic diversity than Cele Black sheep. Our results provide new theoretical support for sheep breeding.

## Figures and Tables

**Figure 1 animals-12-02520-f001:**
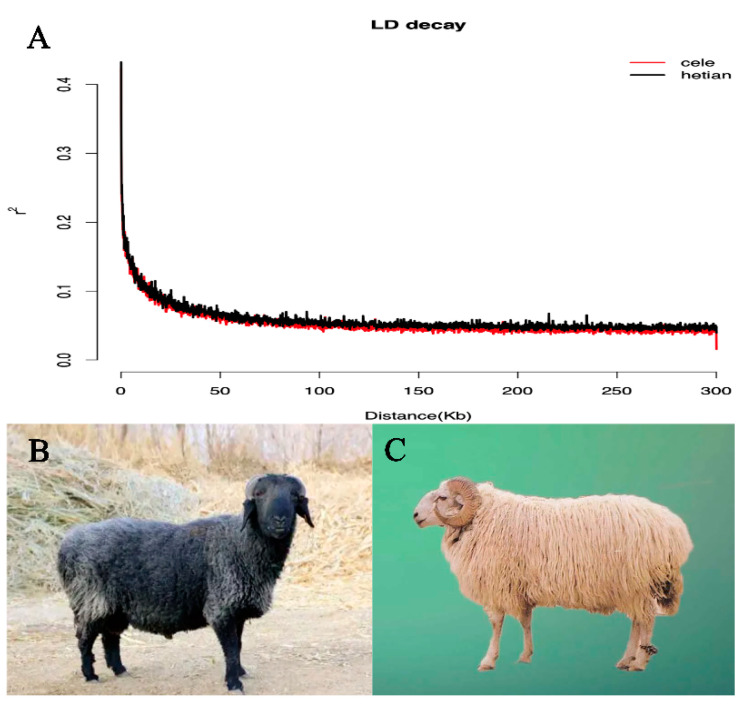
(**A**): The decline curve of linkage disequilibrium (LD) between Hetian and Cele Black sheep. (**B**): Cele Black sheep. (**C**): Hetian sheep.

**Figure 2 animals-12-02520-f002:**
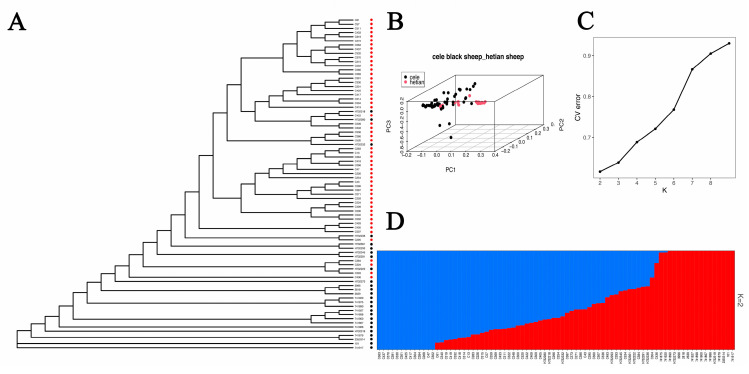
(**A**): Phylogenetic tree: red circles represent Cele Black sheep, black circles represent Hetian sheep. (**B**): Principal component analysis. (**C**): Cross validation errors of ancestral clusters K. (**D**): Population structures with number of ancestral clusters K equaling two. Each color represents one ancestral cluster. The length of colored segments represents corresponding ancestry attributions.

**Figure 3 animals-12-02520-f003:**
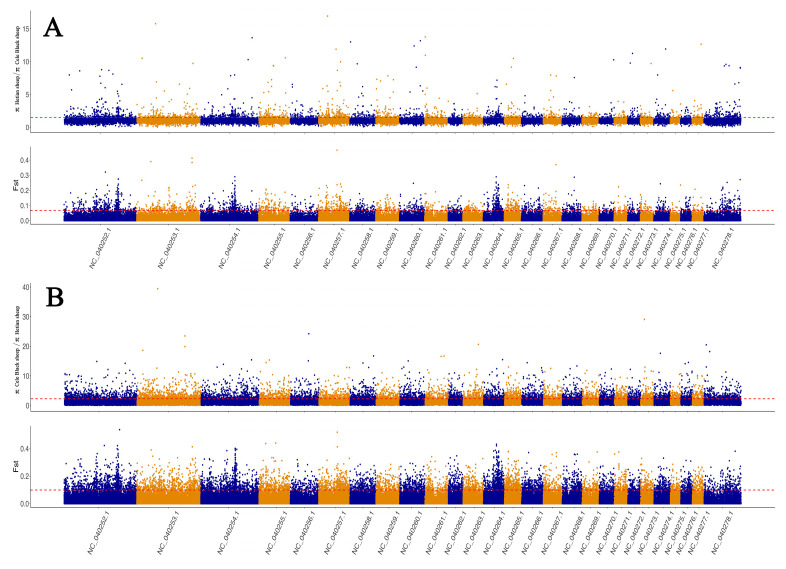
(**A**): The area above the dotted line includes Hetian sheep/Cele Black sheep are subject to selection. (**B**): The area above the dotted line includes Cele Black sheep/Hetian sheep subject to selection.

**Figure 4 animals-12-02520-f004:**
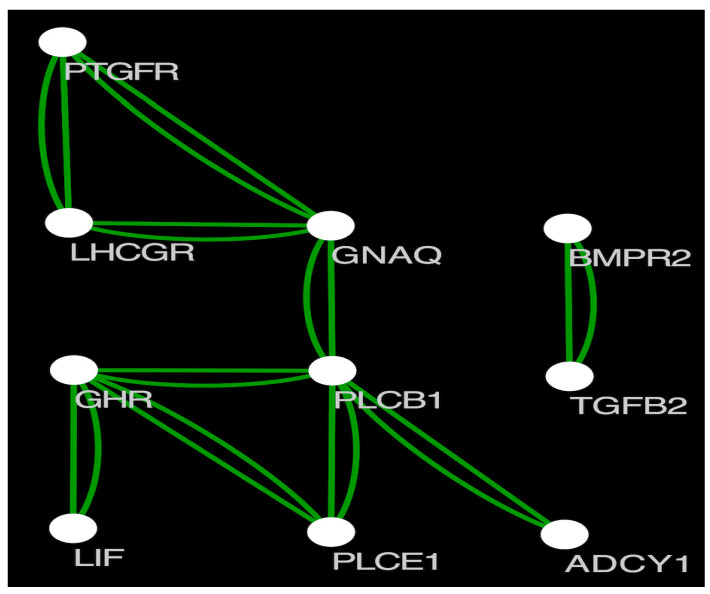
Protein interaction map of the screened genes.

**Table 1 animals-12-02520-t001:** Comparison of genetic parameters between Hetian and Cele Black sheep population groups.

SPECIES	PIC	Ho	GD	π	K
Hetian sheep	0.206261	0.20590242	0.247823937	0.292004	45545.19
Cele Black sheep	0.198109	0.20245098	0.237993324	0.284769	29202.22

## Data Availability

Raw read data were submitted to the NCBI Sequence Read Archive under the accession number PRJNA833897.

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
