# Peer review of "Identification of Signatures of Selection for Litter Size and Pubertal Initiation in Two Sheep Populations"

_animals, 2022, doi:10.3390/ani12192520_

Round 1

Reviewer 1 Report

Dear Authors,

To improve this manuscript, it is recommended to consider the following points:

1-      The introduction is not appropriate to show the importance of the manuscript topic.

2-      The discussion of this manuscript is not enough for this topic. A lot of research has been done on reproductive traits that can improve the discussion of this article.

3-      Some other points and writing errors are mentioned in the enclosed file.

Sincerely

Reviewer 2 Report

Fecundity is an important economic trait of sheep that directly affects their economic and productive efficiency. Authors tried to identify single nucleotide polymorphism (SNP) loci associated with sheep puberty or litter size which could be used in future breeding programs to improve fertility. I raised several doubts and suggestions.

1. The location where the 342 ewes were sampled from should be stated.

2. Authors stated that 54 Cele black sheep (Group CL) and 26 Hetian sheep (Group HT) were used for selection signatures analysis. The litter size of these sheep should be added. Whether the litter size of 54 Cele black sheep was 2? Otherwise, whether it was more reasonable to divide the Cele black sheep to 2 group and reanalyze the data to obtain the genes with litter size?

3. In 2.2 Library construction and sequencing. Authors stated that an aliquot of the product was pooled. Why did you need to pool these product? You did not sequence these sample separately?

4. Figure A1A, Figure A2A, et al. I suggest the authors change to Figure 1A, and Figure 2A, et al.

5. 3.4 Selective signatures analysis. Authors stated that based on GO annotation and KEGG enrichment analysis, genes associated with pubertal initiation or litter size were detected. I think it may be very subjective to distinguish these genes associated with the two related traits?

6. In 3.5 SNP loci validation. The locations of these SNP loci in PAK1 anADCY1 also should be added.

Round 2

Reviewer 1 Report

Dear Authors,

To improve this manuscript, consider the following points.

Sincerely

Correct the instance between the word and reference number in all text. Also correct the following points.

Page 2

between high and low fertility populations in Bamei mutton sheep and found that JUN, ITPR3, PLCB2, HERC5, and KDM4B genes might be associated with polytocous

heat resistance, rough processing resistance, and disease resistance [20].

Page 7

using NJ tree, ADMIXTURE, and PCA.

can be an important basis in for the improvement and development

In our study, the genetic diversity of Hetian sheep

Page 8

This is consistent with previous previously published results [22, 23].

populations are suitable to for the selection signatures

in Kazakh, Chinese Merino, and Hu sheep [50]

Reviewer 2 Report

I am satisfied with the response I concerned.